# A Framework to Incorporate Biological Soil Quality Indicators into Assessing the Sustainability of Territories in the Ecuadorian Amazon

**Thony Huera-Lucero, Juana Labrador-Moreno, José Blanco-Salas \* and Trinidad Ruiz-Téllez**

Department of Vegetal Biology, Ecology and Earth Science, University of Extremadura, Badajoz 06071, Spain; thonyhuera17@gmail.com (T.H.-L.); labrador@unex.es (J.L.-M.); truiz@unex.es (T.R.-T.)
\* Correspondence: blanco_salas@unex.es; Tel.: +34-924-289-300 (ext. 89052)

**Abstract:** In Amazonian Ecuador, land-use change from tropical rainforest to different productive purposes is leading to irreversible situations from an environmental perspective. The objective of this paper was to highlight the significance of the biological components in the soils in Amazonian Ecuador, and the importance of considering biological soil quality indexes when assessing environmental impacts in the soils of tropical Pan-Amazonian areas. Since the literature on the subject is dispersed and inaccessible, a bibliographic review was conducted, with the aim of compiling protocols and proposals for practical utilization. We compiled tables, including specific indicators from the biological point of view. We present the available methods for assessing the sustainability of Amazonian territories through the analysis of soil quality. Our contribution facilitates an edaphic perspective to be taken into account in decision-making processes for sustainable territorial development.

**Keywords:** biological indicators; soil quality; sustainability; tropical soils; Amazon region; Ecuador

## 1. Introduction

In territorial development planning processes, decision-making is an important point to consider, especially in areas where irreversible situations are occurring, as is the case of land-use change from tropical rainforest to different productive purposes in Amazonian Ecuador. Soil is an irreplaceable natural resource, with complex and sensitive functionality; living beings in soil play a fundamental role as indicators of soil status or disturbance. The objective of this paper is to highlight the significance of the biological components in the soils in Amazonian Ecuador, and the importance of considering biological soil quality indexes when assessing environmental impacts in the soils of tropical Pan-Amazonian areas. Since literature on the subject is dispersed and inaccessible, it was decided to address this bibliographic review with the aim of compiling protocols and proposals for practical utilization.

## 2. Methodology

For the purpose of this paper, bibliographic research was carried out using online databases retrieved from the Library Service of the University of Extremadura (Spain). The most commonly consulted scientific databases were: Web of Science, Scopus, Google Scholar, Science Direct, Dialnet, Cyberthesis, and Mendeley. All searches were carried out between September 2019 and January 2020. There was no exclusion of any time period in the search and no language restrictions were applied. The keywords used were "biological indicators" and "soils", "soil quality", "tropical soil", "sustainability" and "soil", "soil biolog\*". Articles before 1999 were only considered if there were no related papers published after the date. Studies that assessed biological soil quality were included, as

well as studies on the sustainability of the Amazon region of Ecuador, regarding the proper use of land and the development of sustainable production systems. A critical reading of full texts and/or complete abstract was made in order to select the most valuable information; part of this appears in the tables and groupings according to the focus of the research. Finally, a synthesis document was elaborated, which was structured as shown in Figure 1.

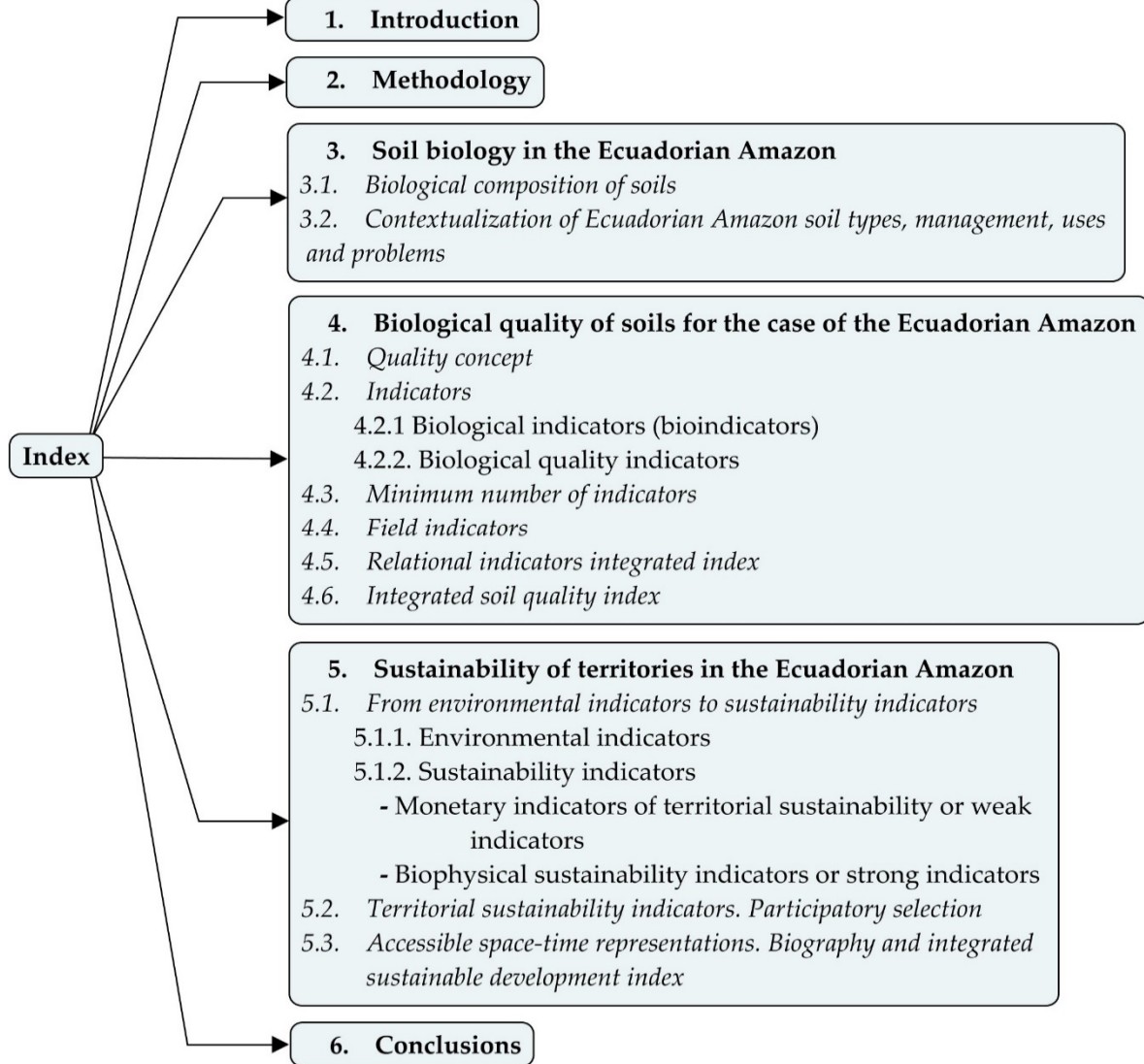

**Figure 1.** Structural index of this review.

## 3. Soil Biology in the Ecuadorian Amazon

### 3.1. Biological Composition of Soils

Soil is a dynamic and non-renewable live system whose condition and operation are crucial for food production and the preservation of environmental quality, and it is a key element to assess sustainability [1–3]. From the biological perspective, soil is a habitat where a large number of organisms reside, grouping in communities of fauna, bacteria, fungi, and algae, which interact with each other, performing important functions [4]. The fauna of the soil includes a variety of organisms of different species and sizes—the most diverse group being arthropods [5]. Depending on their diameter or length, they are usually classified as macrofauna, > 2 mm; mesofauna, 0.1–2 mm; and microfauna, < 0.1 mm [4–7]. The invertebrates that make up the macrofauna participate as engineers in the formation

of the different properties of the soil, in the crushing of plant matter, and some as predators [8]. The abundance, composition, and diversity vary from one land use to another [6]. They depend on soil management [9]. The organisms that make up the mesofauna are called microengineers. They improve the physical properties of the soil and break down organic matter (OM) on a small scale [10]. They participate in the formation of edaphic microstructures as catalysts of microbial activity [7]. Invertebrates are the most abundant component of microfauna, with a high species richness. They are considered the most important predators of bacteria and fungi [4,10].

All of them constitute the soil biota that is summarized in Table 1, which shows a classification according to the size of the main groups and the main functions that they are known to carry out in the soil.

**Table 1.** Organisms that constitute the soil biota and their main functions. Source: own elaboration based on [4,6,7].

| Living Organisms that Make up the Soil Biota | | |
|---|---|---|
| **Group** | **Organism** | **Function** |
| Macrofauna >2 mm | Earthworms | They form a network of tunnels, mixing and digging the soil, producing excreta below and above the ground, modifying the water and chemical properties [4]. They transform the physical properties of the soil, benefiting the formation of aggregates, the movement and retention of water, as well as the gas exchange [8]. They act as ecosystem engineers in pore formation, water infiltration and OM humification and mineralization [6,10]. |
| | Beetles | They participate in litter fractionation and in OM decomposition and mineralization processes [6]. The conservation of the coleoptera family can be a support for possible evaluations of the environmental quality [11]. |
| | Ants | Their mounds are rich in nutrients such as N, P, K, Ca, Mg, and Fe, favoring the proliferation of microflora and micromesofauna [12]. They affect the soil structure, mixing the horizons of the profile and recycling part of the elements that leach from the surface [13]. They modify the physical and chemical properties of the soil [10], and contribute to the formation of aggregates, water filtration, and aeration [8]. |
| | Termites | They contribute to bioturbation processes [4], and intervene in the crushing of plant remains and in the decomposition of woody material [6]. |
| | Snails and slugs | They participate in litter fractionation and in OM decomposition and mineralization processes [6], and the fragmentation of leaf litter, when they mobilize they secrete mucus, increasing area for microflora activity [4,8]. Their mucus helps aggregate formation, improving soil structure and properties [10]. |
| | Centipedes and Millipedes | They live among the leaf litter or under the bark of trees and rocks, they play an important role as predators, and others participate in leaf litter fragmentation, speeding up the OM decomposition process [8,10]. |
| | Enquitraeid worms | They participate in leaf litter maceration and plant remains, facilitating the transport of excavators, and they can also act as predators [10]. |

**Table 1.** *Cont.*

| Living Organisms that Make up the Soil Biota | | |
|---|---|---|
| **Group** | **Organism** | **Function** |
| **Mesofauna 1–2 mm** | Collembola | They are decisive in the recycling of organic waste, dividing and crushing them, their excreta benefits the roots by the continuous release of nutrients. They participate as predators of nematodes and fungi [7]. They are considered a decisive element in the recycling of organic remains, and contribute to the structure of the soil [10]. |
| | Mites | Their role is to fragment leaves and dead wood, disperse microbial and fungal spores in the soil. Some species are predators of other microarthropods, nematodes, and mites [10]. Moreover, they contribute to the soil stability and fertility [7], and in OM decomposition [14]. |
| | Nematodes | They are concentrated in the roots, serve as food for plants, do not participate in OM decomposition [10], reflect the OM availability in different ecosystems, are the link in the food chain between microorganisms and complex organisms [7]. Some can resist soil disturbances and chemical pollutants [15], others are parasites. There are mycophagus, bacteriophages participate in the regulation of available nitrogen and phosphorus and influence the Rhizobium nodulation [10,14]. They are important agents of the nutrient cycle and regulators of soil fertility, and they work as biological control agents [15,16]. |
| | Protura, diplura, and pauropoda | They inhabit deep strata, under trunks or stones, they are detritivorous and depend on moderate and constant humidity, they consume microorganisms and fungal hyphae, which is why they are considered to be involved in decomposition, some of their representatives are predators and phytophagous [7,10]. |
| **Microfauna <2 mm** | Protozoa | They are considered the most important predators of bacteria and fungi. Moreover, they regulate microbial communities, and as pathogenic insects, represent an important biological control [4,10]. |
| **Fungi** | | They are involved in processes of decomposition, mineralization, and cycling of nutrients [2,17]. By forming symbiotic associations, they increase the efficiency of plants to absorb nutrients [4], increase soil aggregation and participate in the carbon cycle [17], and allow plants to survive and efficiently absorb phosphorus from the soil [18]. They improve the soil health and plant species growth, provide greater absorption of nutrients, uptake of immobile ions, tolerance to toxic metals, root pathogens, and unfavorable conditions for plants in tropical ecosystems [2,10,19]. |
| **Algae** | | They are photosynthesizing organisms involved in primary production, organic carbon (OC) compounds, and soil structure [10], and are colonizers. In association with fungi, they form lichens and contribute to soil formation, degrading minerals or rocks by excreting organic acids [20]. From the production of carbohydrates, they form soil aggregates and stop their erosion [21]. Given the variable and morphologically similar nature of the majority, today, they are identified using molecular techniques [22–24]. |

**Table 1.** *Cont.*

| Living Organisms that Make up the Soil Biota | | |
|---|---|---|
| Group | Organism | Function |
| | Bacteria | They rarely contribute to biological activity. They can be considered bags full of enzymes [10]. In the soil they are very numerous and genetically different. Some degrade chemical compounds and others form nodules in the roots of legumes, with the function of fixing atmospheric nitrogen through heterocysts. Cases such as Pseudomonas can be pathogenic [10,25]. There are cyanobacteria (photosynthesizers and autotrophs) [26]; Actinobacteria are colonies similar to fungal mycelia, like Actinomycetes that degrade OM to form humus and participate in the mineralization process, others can fix N or regulate the composition of the microbial community in the soil. They secrete enzymes that serve for the biological control of nematodes, insects, and other soil organisms. Their number on agricultural land is high [2]. |

### 3.2. Contextualization of Ecuadorian Amazon Soil Types, Management, Uses, and Problems

The Ecuadorian Amazon constitutes 45% of the National territory, occupying 115,613 km$^2$, extending from the Andean mountain foothills, with the appearance of transitional forests at 1300 m.a.s.l., to the east of the Amazonian plain, forming less than 2% of the Amazon River basin. The average rainfall fluctuates between 2000 and 5000 mm annually, with a temperature of 24 °C during the year, which constitutes a warm damp climate [27,28].

The Ecuadorian Amazon has been referred to as "the most important source of fresh water and biodiversity" for its global climate regulatory function as a greenhouse gas sink [29]. It is made up of six provinces: Sucumbíos, Orellana, Napo, Pastaza, Morona Santiago, and Zamora Chinchipe (Figure 2), which have a variety of soil types, from the Andes to soil derived from volcanic ash with specific characteristics [27,30,31]. Due to the geological and influence of meteorological factors, poorly developed soils appear with acidic pH, that are susceptible to erosion due to high rainfall, low fertility and nutrient availability, the presence of toxic aluminum for plants, and high moisture retention [27,31,32].

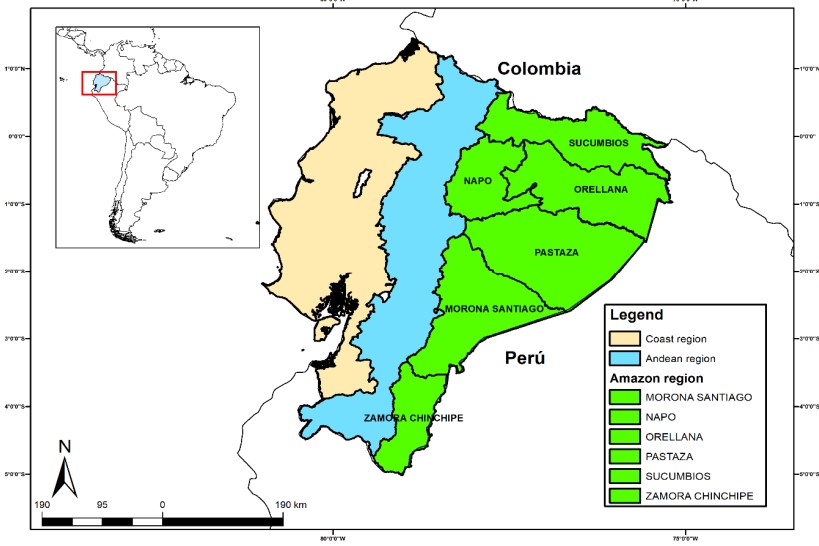

**Figure 2.** Political map of the Amazon region of Ecuador.

According to the American Soil Taxonomy (USDA) system, the soils present in the Ecuadorian Amazon belong to the typological units that are collected in Table 2, prepared from information compiled by the SIGAGRO (Geographic and Agricultural Information System) of the Ministry of Agriculture Livestock Aquaculture and Fisheries of Ecuador [31–33]. The most representative soils of the Amazon territory are those of order inceptisol, entisol, histosol, and mollisol [34,35].

**Table 2.** Taxonomic classification of soils in the Ecuadorian Amazon, based on [31–33].

| Taxonomic Description of the Soils of the Ecuadorian Amazon | |
|---|---|
| **Order** | **Suborder** |
| **Alfisol**: Mineral soils developed in stable reliefs of erosive tectonic origin, with clays in the surface horizon, and with more than 35% base saturation; because of the natural reserve of nutrients, they are considered good for short-cycle farms and forages. | |
| **Entisol**: Soils with the scarce formation of horizons, either due to susceptibility to erosion and flooding, or due to the short evolution time; of variable fertility, due to the original material dragged by alluviums or rivers; present agricultural vocation, but the excess of humidity is their main limitation. | Typic Udifluvents |
| **Histosol**: Organic soils that are formed when the OM exceeds its mineralization, that are without Indian origins, and are limited in swampy areas that are poorly drained under conditions of continuous saturation that prevents the circulation of oxygen. In this type of soil, sustainable use is limited to forested formations, grasslands, and in some cases, intensive crops. Their ecological importance is because they act as carbon sinks due to the speed of OM contributions they receive. | Aquept Fibrist |
| **Inceptisol**: Soils with a lack of edaphological maturity, whose genesis is of rapid formation. They occupy areas of irregular topography, with both humid and subhumid climates, and have variable chemical and physical properties, from acid to alkaline, sandy to clay. They have—more or less—a base saturation of 60%, and have been used by the agricultural sector for key crops for the economy. They are represented by a forest cover, pastures, and representative crops. | Andept Tropept |
| **Mollisol**: Very fertile soils, with a surface horizon rich in OM and more than 50% of base saturation; occupy structural reliefs, with slight slopes towards the coast and sedimentary valleys of volcanic origin; develop in a variety of climatic regimes, mostly grassland vegetation, with high agricultural yield, sometimes the highest in the world. | Udoll |
| **Oxisol**: Mineral soils with low fertility, from marginal slopes of the Real and Cutucú mountain range; occupy tropical areas that have gone through prolonged processes of weathering and washing due to heavy rainfall; have red, yellowish colors due to iron accumulation, oxides, and hydroxides of aluminum. Their main use is for livestock, followed by forest areas, but they are not suitable for agricultural activities. | Ortox |
| **Ultisol**: Acidic soils, which are products of chemical weathering, from eastern and western Andean hills, present in any humidity regime, with a base saturation of less than 35% providing acidity. These soils are characteristic of humid forests that are part of conservation areas. | |

Due to the biological richness and endemism of the Ecuadorian Amazon, it is one of the largest ecological reserves, with a high potential to provide ecosystem services to local populations. Being subject to high rates of deforestation and changes in use affects the biodiversity, soil, and water, as basic components of an ecosystem, causing the decrease or loss of possible ecosystem goods and services [29,36].

In tropical regions, the change from forests to pastures and agricultural production, on the one hand, leads to losses in terms of fertility, quality, and biodiversity in the soils, and on the other hand, implies the addition of nutrients and amendments to meet the demand for crops; hence, to promote a change in the productive matrix with a sustainable vision, the best strategy is the implementation of agroforestry or silvopastoral systems [2,30,36,37]. This confirms that the Ecuadorian Amazon is not suitable for intensive agricultural activity, but for productive systems similar to the forest [31]. The most common systems are described below.

**Silvopastoral systems:** Due to the dependence of part of the Amazonian population on livestock activity, it has been necessary to generate production models similar to the forest (silvopastoral systems) [34]. This consists of combining the use of trees, pastures, and animals to increase production, which at the same time brings other benefits, such as wood, biodiversity conservation, carbon fixation, and the protection of water basins and soil [27]. These systems represent a sustainable alternative, which responds satisfactorily to the socio-economic reality of tropical countries, mainly with regard to food demand [38]. In the Ecuadorian Amazon, with colonization, the forests were intervened and used for timber resources, and the establishment of pastures and small orchards [37]. Livestock has turned soils into degraded lands, dependent on mechanization and agrochemicals [35]. Despite this problem, there is an interest in understanding the state of those soils, and implementing strategies or systems compatible with the environment, soil, edaphic life, and livestock farmers [3]. The adoption of silvopastoral systems has increased the biomass in the soil surface, the level of OM in the surface horizon, and thereby the availability and recycling of nutrients, hence improving the soil structural index [1,27,30]. On the other hand, the grass influences the system; for example, long-cycle gramalote grass (*Axonopus scoparius*) does not affect the soil, but forms associations with short-cycle pastures, such as honey grass (*Setaria plendida*), Amazonian kikuyo (*Brachiaria humidicola*), dallis (*Brachiaria decumbens*), or elephant grass (*Pennisetum purpureum*), in which they can produce important erosion and soil degradation processes [1].

**Crops:** In the Ecuadorian Amazon, many agricultural projects have been promoted with African palm, coffee, cacao, rice, and grassland, from which extensive (monoculture) crops have originated that involve the deforestation of large-scale virgin forest. Projects that expanded the agricultural frontier for state support (credits) [34]. Incentives for planting African palm and palmito, mainly, have caused significant losses in biological biodiversity and vegetation, altering the multifunctionality that ecosystems play [7,34]. The management techniques used in monocultures affect the soil community and make them dependent on external inputs [7], without neglecting the deterioration and degradation of soils that are of little interest to large producers, whose only interest lies in the profitability and economic benefits that they can obtain.

On the other hand, there are polycultures or chakras, which have developed Amazonian populations for family subsistence. The chakras are implanted within the forests, occupying small areas, and include some natural plant species. In addition, they open a variety of crops, such as cassava, banana, Chinese potato, naranjilla, sugarcane, as well as fruit trees of chontaduro, guaba, papaya, and wild grapes, naturally [37], without neglecting the variety of medicinal plants. The ideology is to maintain a balance between the chakras and nature without altering the structure, the life that inhabits the area, or the soil.

**Agroforestry systems:** In Amazonian ecosystems, they fulfil important functions. These systems improve the relationships between the soil, water, and air components of the ecosystem [36,39]. When linking crops, such as cocoa or coffee, the damage caused to the soil is minimal. The constant addition of litter and root development that increase the soil's OM produces conditions similar to those of a natural forest [1,27]. The similarity of conditions of these systems with forests allows the development of processes, such as recycling, fixation, and mineralization of nutrients [36]. They prevent the implantation of monocultures and the impacts of deforestation. At the same time, they generate other economic inputs for producers, such as wood production, firewood, wild fruits; together with environmental services, such as the reduction of greenhouse gases, carbon sequestration, and protection of natural resources (soil, air, water) [38]. They serve as habitats for plant and animal species, from microorganisms, vertebrates, and amphibians of the soil, to large mammals and birds [37,39].

In Ecuador and tropical regions in general, the conversion of forests to agricultural and livestock systems is the most important reason for land-use change [40]. These changes lead to the alteration and loss of biodiversity, and mainly to deterioration of the soil [27,31].

**Forests:** The Amazon is characterized by its lush vegetation, and for having soils with a superficial horizon rich in OM. These data led us to mistakenly think that they were fertile lands. After several

investigations in deforested areas, it has been proven that Amazonian lands are not very fertile. They become impoverished quickly in the absence of organic remains provided by the vegetation cover [39]. In Amazonian soils, despite the low availability of nutrients, forests have adapted to weathering and washing conditions, capturing the nutrients generated by the decomposition of the OM directly with their surface roots [34].

In Ecuador, oil activity is indirectly responsible for the current state of Amazonian forests [41]. With the appearance of oil companies, agricultural and livestock activities have positioned themselves as being the main and majority use of the Amazon region [37,42]. Currently, institutions and research centers are developing projects aimed at the sustainable use of natural resources [39]. These projects are associated with environmentally friendly management systems and practices that benefit the Amazonian community. In the Ecuadorian Amazon, most forests that are not part of the National System of Protected Areas of Ecuador (SNAP) or some protective forest are degraded either by the selective extraction of species or because they are secondary forests [42].

## 4. Biological Quality of Soils for the Case of the Ecuadorian Amazon

*4.1. Quality Concept*

Soil, water, and air quality are the components of environmental quality, but if we focus explicitly on soil resources, the quality is more complex to define due to its variety of components [43], and the enormous amount of interrelationships between its components. Chemical, physical, and biological parameters can be analyzed and integrated to form a soil quality index that allows comparisons between different uses or management practices [44]. Soil quality is usually focused on agricultural production [45], but is also a critical component in the maintenance of sustainability [38], and human and environmental health [44]. A soil's quality is defined "as its ability to function within an ecosystem; to sustain or improve animal or plant productivity; to maintain and control environmental quality, and to support the habitability and health of man" [46].

The biological component is of great importance in assessing the management of land uses. This allows the implementation of agro-ecological management that favors agricultural production and biodiversity [38]. Some authors consider that for a soil to be considered high quality, it must meet criteria related to respiration, biomass, and its microbial activity [47], which correspond to biological parameters.

From a general point of view, the quality of the soils of the Ecuadorian Amazon is marked by fine clay textures, with good granular structure, on the surface horizon [36]. It has high OM content of low quality, low fertility, and acidic pH that limit the availability of nutrients, such as phosphorus and leaching of changeable bases (potassium, calcium, and magnesium), limiting its use [39,48,49]. A very thin superficial horizon with intense biological activity due to the accumulation of OM, and the presence of humidity, influences biogeochemical behavior [27]: phosphorus deficiency, the presence of sulfur, changeable bases [27,36], and high levels of iron and aluminum fixation [50]. As phosphorus is a critical macronutrient [19], low levels of native phosphorus represent one of the biggest obstacles to food production. In general, the dark colors on the ground are associated with OM and high biological activity [49], and reddish colors are associated with ferric minerals [1,48]; a characteristic behavior of Amazonian soils.

Consequently, when talking about soil quality, the most important thing is to know if the focus is from the point of view of agricultural productivity, or environmental or human health. In addition, the quality can refer to the physical, chemical, or biological components of the soil. It is much more frequent that indicators are considered for the evaluation of the quality of the soil. In the context of the Amazon, biological indicators or bioindicators are very useful.

*4.2. Indicators*

An indicator is a parameter that allows for the verification of the soil's situation in relation to its state of conservation, pollution, productivity, or any other characteristic that provides information regarding its current and potential status [9,39,43]. These indicators are classified into four categories: visual, physical, chemical, and biological indicators to assess the quality of a soil [38].

Visual indicators are obtained with field visits, farmers' perceptions, and local knowledge. These are based on observations and interpretations, such as the exposure of the subsoil, the color of the soil, the presence of gullies and weeds, the flooding, runoff, or poor vegetation development—all of these aspects are indications of alterations in soil quality [43,51,52]. Physical indicators are related to the structure of the soil, as is the case for porosity, bulk density, penetration resistance, water retention capacity, hydraulic conductivity, aggregate size, depth, and texture. These mainly reflect the limitations of root growth, seedling sprouting, infiltration, or movement of water within the soil profile, transfer and cycling of nutrients [38,51]. Chemical indicators include soil–plant properties, such as water quality and the availability of water, and nutrients for plants and microorganisms. Among the most common are pH, electrical conductivity, organic matter content, cation exchange capacity, and nutrients (total N, total phosphorus and potassium) [38,43,50].

Finally, biological indicators are related to the decomposition and incorporation of animal and plant residues in the soil by living organisms, controlling the supply of nutrients and humus to the ecosystem [53]. These indicators are based on soil respiration, microbial biomass, the amount of species and groups of edaphic fauna, as well as tests on enzymatic activities [54]. They act as early (microbiological and biochemical) signals of soil degradation or improvement due to their sensitivity [38,53]. The close relationship of the quality of the soil with the functions developed by the edaphic life has made them valuable indicators of disturbance, based on both their functions and their diversity, density, and abundance [38]. When the indicator is a living being, it is called a biological indicator, or bioindicator.

4.2.1. Biological Indicators (Bioindicators)

An edaphic bioindicator is every living being that responds easily to external (soil) stimuli through changes at the organism level [47]. Bioindicators need to belong to large, diverse taxonomic groups, of wide geographical and ecological distribution [6,27]. They must be easy to handle, visible at any time of the year, with easy reproduction, and be abundant and preferably sedentary. Depending on the presence/absence of changes against these soil variations (stimuli), bioindicators are called sensitive or tolerant [6,12,39].

We present below the most representative organisms of soil biota, considered to be bioindicators, with a special focus on the context of the Ecuadorian Amazon described in Section 3.1. The information and descriptions as functions, characteristics, and functionalities of each group are summarized as Supplementary Material S1. Table 3 shows the applicability of some more representative organisms as edaphic bioindicators, depending on their abundance or absence.

4.2.2. Biological Quality Indicators

These are parameters that serve to evaluate processes carried out by living beings in the soil, such as the transfer of nutrients from the soil to the plant, the dissolution of minerals that live in the mother rock, the mineralization of the OM, the stabilization of the soil structure that produces the OM, the cohesion of aggregates, and/or the formation of galleries that aerate and give porosity to the soil. As long as they can be measured, they can provide information on the condition and operation of the soil [38]. Table 4 summarizes the most important indicators, which we describe below.

**Table 3.** Organisms used as edaphic bioindicators.

| Organisms of Soil Biota Considered Edaphic Bioindicators | | |
|---|---|---|
| **Organism** | **Indicator** | **Author** |
| **Earthworms** | They are recognized for presenting sensitivity to anthropogenic disturbance, proposing them as indicators of soil degradation [4,13]. Several authors propose them as a biological indicator of the state of conservation/alteration of the soil according to the composition and abundance [6]. Their presence indicates preserved habitats [55]. | |
| **Beetles** | They are considered excellent bioindicators to evaluate anthropogenic intervention due to the high sensitivity to environmental variations and deterioration of ecosystems [6,40]. According to the ecological niche they occupy, they are considered as indicators of the conservation status of the ecosystem [39]. | |
| **Termites** | The presence of termites indicates less conserved habitats or habitats with a certain level of degradation, considered opportunistic organisms due to resistance to induced disturbances [6]. On the other hand, they are potentially the most important taxa as ecological indicators, because they are at the ecological center of many tropical ecosystems [12], and moreover, for their sensitivity to environmental or anthropogenic disturbances in biotic systems [56]. | |
| **Snails and Slugs** | Used to indicate the state of disturbance in the edaphic environment, they are very sensitive to sudden changes in humidity and temperature, associated with vegetation cover and the entry of residues [6]. For this reason, they are considered indicators of humidity and temperature changes. | |
| **Centipedes and Millipedes** | Used to indicate the state of disturbance in the edaphic environment, they are very sensitive to sudden changes in humidity and temperature, associated with vegetation cover and the entry of residues [6], and because these changes can influence its functions and abundance [8]. | |
| **Enquitraeid worms** | They are drought-sensitive organisms [10]; for this reason they are considered drought indicators. They can be considered bioindicators of soil stability and fertility [7]. | |
| **Collembola** | Due to their action of reducing fungal concentrations, in crops they are used as bioindicators of soil contamination, since they have whitish and soft bodies, they are considered an indicator group of fertility and stability of the edaphic environment due to their sensitivity to chemical products and environmental disturbances [7]. For the changes in their composition, they are considered indicators of ecological variations, due to the influence of agricultural practices, making the presence of taxa effective as bioindicators of herbicide treatment [57]. | |

**Table 3.** *Cont.*

| Organisms of Soil Biota Considered Edaphic Bioindicators | | |
|---|---|---|
| **Organism** | | **Indicator** | **Author** |

| Organism | | Indicator |
|---|---|---|
| **Mites** | **Oribatida Uropodinos** | Due to their morphological and bioecological characteristics, they are very demanding in terms of habitat quality, suggesting them as potential bioindicators of disturbance, as they are sensitive to OM content, humidity, pH, agricultural practices, use of insecticides, and environmental changes. They respond positively to good soil aeration conditions, considering them indicators of stable and productive soils, and in soils not intervened as bioindicators of low heavy metal values [7]. |
| | **Astigmata** | Surviving unfavorable environmental conditions, they are proposed as good indicators of disturbed soils [7]. |
| | **Gamasinos** | Biological indicators of soil stability and fertility; due to their susceptibility to environmental disturbances and the fragility of their whitish bodies, these characteristics also make them a good indicator of soil quality, since they are abundant in the least disturbed [7]. |
| | **Prostigmata** | When they have high dominance, it is considered as an indicator group of the aridity and the imbalance of the edaphic communities is irreversible, because they have a high reproductive potential, which allows them to adapt to the disturbance and for this reason they are considered disturbance indicators [7]. |
| **Nematodes** | | They act as biological control agents for pests and insects, qualifying them as powerful bioindicators of ecological conditions [15]. Through appropriate analysis of the nematode community, the level of contaminant disturbance and changes in land use can be estimated [43], therefore, they are considered indicators of sensitivity and stability [16]. |
| **Protura, Diplura and Pauropoda** | | Due to their morphology and trophic functions, they are considered indicators, they are very sensitive to agricultural practices, thereby reducing their population [7]. |
| **Arbuscular mycorrhizal fungi** | | The mycorrhizal association has recently been seen as an important indicator to assess soil quality. They also represent a key group of organisms in the soil that can affect plant productivity, biodiversity, and characteristics related to ecosystem sustainability [17]. Moreover, they are considered bioindicators of soils contaminated by heavy metals [58]. |
| **Algae** | | Excretions of fatty acids and carbohydrates, they stop erosion-forming aggregates [10]. Due to their nature and similar morphology, molecular techniques are used for better identification [22,24]. |
| **Bacteria** | | The actinomycetes in tropical soils are one of the most important bacterial groups [2], as indicated by a recent review of soil bacteria worldwide [59]. Their systematic classification is based on molecular techniques (16S rRNA sequencing) of soil microorganisms [26,59]. |

**Table 4.** Biological quality indicators for Amazonian soils.

| Indicators | Methodology | What Do They Indicate | Author |
|---|---|---|---|
| **Organic matter** | Wet oxidation method with modified Walkley–Black dichromate | OM is considered an important indicator of soil quality and productivity [30], because it influences a wide range of soil properties. On the other hand, it is the most important component of soils, since it plays a key role in determining physical, chemical, and biological processes, exercising crop production [47]. OM is a globally recognized variable as the universal indicator of soil quality [53]. It is considered a sensitive indicator to changes due to soil management [58,60]. | |
| **Particulate organic matter** | Modification of physical fractionation | Both are positioned within the most sensitive indicators, helping to identify changes manifested at different depths and in the face of management practices [53]. Organic phosphorus allows the prediction of nutrient availability in the short term [19]. | |
| **Organic phosphorus** | 1970 Dewis and Freitas Method | | |
| **Potentially mineralizable nitrogen** | Method outlined (Keeney and Nelson 1982) Modified Waring and Bremner Method (Keeney 1982) | It is a necessary indicator for a complete evaluation of the soil, it is associated with the quality of the OM. On the other hand, it corresponds to the amount of organic soil nitrogen that can be converted by microbial activity to soluble inorganic forms and due to its sensitivity it can be used as an indicator of the production capacity of the soil or as an indicator of the nitrogen contribution of the soil to support recommendations for the application of N [53]. Moreover, it is an indicator highly sensitive to changes in use in subtropical soils. | |
| **Microbial biomass** | | Microbial biomass is considered an indicator of soil fertility and quality, and is negatively affected by changes in land use and agricultural practices [60]. Furthermore, they are used as indicators of the first environmental changes by deforestation. | |
| **Carbon of microbial biomass** | Fumigation-extraction method with chloroform (Jenkinson and Powlson 1976) Substrate-induced breathing method (Anderson and Domsch 1978) | Microbiological parameters that have been used as indicators of the effect of agricultural practices and pollutants on soil quality [47], and as indicators of the relationship between biota and the restoration of degraded systems, allowing us to know the abundance and population structure of microorganisms [58,61]. They have been proposed as indicators of soil quality in natural and agricultural systems, due to the role of microorganisms in the C, N cycle [62], and their sensitivity [30]. In subtropical soils they can serve as potential biological indicators of ecological changes resulting from land use and management practices [62]. | |
| **Nitrogen from microbial biomass** | | | |

**Table 4.** *Cont.*

| Indicators | Methodology | What Do They Indicate | Author |
|---|---|---|---|
| **Soil respiration** | Static incubation, Alkali-trap method (Anderson 1982) | It is carried out by microorganisms under aerobic conditions, it is a useful index to know the amount of easily mineralizable substrate, by determining the amount of $CO_2$ released by the action of biological activity and the easily mineralizable OC [47,54], allowing to evaluate the type of management to which a soil is subjected [47]. Indicator highly sensitive to changes in the use of subtropical soils [53]. Used as an indicator of soil quality and microbial activity, it can also indicate the amount of easily mineralizable substrates [63]. | |
| **Metabolic or microbial ratio** | | The metabolic processes that occur in the soil can serve as early and sensitive indicators against the changes caused by different soil management [53]. This quotient is a useful indicator to monitor changes in OM and is often used as a sensitive index to measure changes in soil OC. Its increase is considered as an indicator of environmental stress after the conversion of forests to farmland [60]. Indicator of availability and quality of microbes, it is also sensitive to other factors, such as the proportion of fungal and bacterial biomass [63]. The stress of the microbial communities can be quantified by means of this parameter that reflects the energy requirement or indicates a change in the bacterial-fungal ratio. At the same time, it could be a useful parameter in the study of bioenergetic changes in developing ecosystems [64]. | |
| **Geometric measurement of enzymatic activity (GMEa)** | $GMEa = (enzyme \times enzyme \times n \dots)^{1/\#enzymes}$ It consists of multiplying the values of each enzymatic activity to know GMEa. | It is a measure of enzyme activity that is proposed as an indicator of recovery in the presence of bioavailable heavy metals [54]. It is an indicator of changes in soil quality under different agricultural management practices and is used to assess the effects of cultivation on soil quality [60]. It has been shown to be a good index (condensing the set of enzyme values) to estimate the quality of the soil, since it is related to other physicochemical or biological properties of the soil. Furthermore, it is an early indicator of change in soil quality and is sensitive to metal contamination [63]. | |
| **Molecular markers, quantitative and real-time PCR (polymerase chain reaction)** | Bead-beating method | The estimation by direct or indirect extraction of nucleic acids (DNA and RNA) from the soil and their subsequent study through molecular biology techniques, such as PCR, has been used successfully, and they have been proposed as indicators of microbial biomass activity [47]. This technique is capable of discriminating between bacterial and fungal biomass. They are the preferred way to assess the structure and dynamics of the soil microbiological community, since microbiological life is a favorable indicator of adequate soil characteristics [65]. | |

**Table 4.** *Cont.*

| Indicators | Methodology | What Do They Indicate | Author |
|---|---|---|---|
| **Enzymatic activity** | Fluorogenically labeled substrates, (Tabatabai 1982) | They are considered as sensitive indicators between reforestation methods and changes produced by agricultural activity [30]. Due to their sensitivity, relationship with biological activity, and rapid response to changes that occur in the soil (use and management), they have been proposed as potential indicators of soil quality [47,62]. Phosphatase activity is a soil indicator to estimate the potential mineralization of organic phosphate [47]. They are used as indicators of changes in soil microbial activity, in response to heavy metals and corrective measures [54], and they show sensitivity to changes in land use and pollution. They have been used as indicators of soil fertility and quality, but while being negatively affected by changes in land use and agricultural practices are proposed as indicators of soil degradation [60,63]. They can also indicate accelerated decomposition [63]. | |

Organic matter (OM): the availability of OM is one of the main components of the soil. It is directly related to the different properties [36,47], such as the influence of temperature and humidity, which condition the mineralization in the microbial phase of the soil [30].

Organic carbon (OC) largely depends on the availability of OM and land use [3,60]. It is part of the different soil processes and is a source of food for edaphic organisms [3]. In pastures and crops, OC decreases by 65% compared to the forest [60], due to the low production of OM. It is among the five best carbon sinks [47].

Particulate organic matter refers to the youngest and most active portion of the OM. It is a reservoir of nutrients for the flora and fauna of the soil. It acts by increasing water carrying capacity and stabilizing aggregates. An analysis of this parameter allows the prediction of short-term nutrient availability [50,53].

Organic phosphorus is an important macronutrient for the functionality of plants. In tropical areas, availability limits plant growth [19], being one of the most sensitive nutrients in tropical soils [53].

Potentially mineralizable nitrogen is the amount of organic N in the soil that is transformed into soluble inorganic forms, such as $NH_4^+$ and $NO_3^-$, by microbial action. It is directly associated with the availability of OM and for its sensitivity it is considered as an indicator of nitrogen production or nitrogen contribution of a soil [53].

Microbial biomass (MB) is determined by the quantity and quality of OM that, at the same time, depends on the use of soil. Microbial characteristics are considered quality indicators [62]. Therefore, when MB is high, it indicates microbial diversity and an optimal environment. However, if they are at low levels, it is a sign of some kind of pollution [62] or due to changes in land use [61].

Carbon from microbial biomass (CMB) is related to the addition of OM to the soil. It indicates the biochemical and microbiological activity of soils. When it is high, it is considered an indicator of soil fertility [3]. It provides knowledge on the abundance and population structure of microorganisms, and is obtained from the difference between samples with C extracted and samples without C extracted [62], according to the fumigation-extraction method with chloroform. CMB is considered an indicator of soil quality [47,60].

Nitrogen from the microbial biomass (NMB), as well as CMB, depends on the MB and the amount of available OM [60]. It is obtained from the difference between fumigated (N extracted) and non-fumigated samples [62], according to the method of fumigation-extraction with chloroform. Actual NMB scores, similarly to CMB, are determined by the conversion factor (mineralized fraction for C and N) [61,62], applied to the general formula that determines the MB.

Soil respiration refers to the production of $CO_2$ as a result of microbial activity, roots, and macro and micro fauna. It is measured under anaerobic conditions and provides information on mineralizable substrates [47]. The larger the population, the greater the amount of $CO_2$. It also relates to the size of plant waste, litter, and biota in general [53], considered as an index of biological activity.

Metabolic or microbial ratio is the index of the relationship between growth and state of latency of BM. It measures the microbial change of the soil with respect to environmental limitations due to changes in use. An increase indicates unfavorable conditions for soil microbes (microbial stress) [60,64].

Geometric measurement of enzymatic activity (GMEa) is a common index to integrate data and information from various enzymes [54] and know the meaning of the enzymatic activity of a soil. Some authors [54,60,63] consider it a good index to estimate the quality of the soil. It can be related to physicochemical and biological properties.

Molecular markers—molecular techniques are capable of differentiating fungal microbial biomass, according to primers that are designed from the 16S and 18S rDNA genes [47]. The genomic study of soils from DNA shows the genetic potential to produce certain enzymes, motivating the study of RNA. These techniques show real information on the state of the soil and the environmental conditions to which microorganisms are subjected [47,65]. The genome of bacteria provides signals when there is some kind of impact on the soil. Currently, the determination of indicators of soil quality is based on

the DNA and RNA of the species of soil organisms. It has great potential, speed, and provides more informative measurements of biota [43].

Enzymatic activity: Table 5 synthesizes some types of enzymatic activity that can be evaluated in soil samples. Their activity is affected by the change of land use, especially in the surface layer [47]. Low concentrations of enzyme activity indicate inactivity of microorganisms. High concentrations indicate the high decomposition of the OM and microbial activity [62]. In tropical soils, the marked variations are probably due to acidity [3], which are considered land degradation indicators, after deforestation [30,60], specifically, fertility and quality indicators [60,63]. Some research in Amazonian soils considers that enzymatic activity indicates ecological changes resulting from land use [62]. According to [54], these activities (Table 5) are the most sensitive indicators to assess the effects of restoration practices and effects of land use change.

**Table 5.** Types of enzymatic activity that can be evaluated in soil samples.

| Enzyme | Substratum | Description | Author |
|---|---|---|---|
| **Dehydrogenase** | 2-*p*-iodophenyl-3 *p*-nitrophenyl-5 tetrazolium chloride | Measures total oxidative activity of the microflora and estimates the microbial activity. Indicates the redox potential and oxidative capacity of the soil. It is proposed as an indicator of microbial activity. Its decrease may indicate the presence of herbicides. | [30,47] |
| *O*-**diphenoloxidase** | | Catalyzes oxidation of phenolic compounds and participates in the formation of humic substances. Degrades recalcitrant organic compounds. | [47] |
| *B*-**glucosidase** | *p*-nitrophenyl *B-D*-glucosidase | Catalyzes hydrolytic processes during the decomposition of OM, the soil predominates, so it is used to study the C cycle, as it degrades cellulose. Indicates presence or absence of herbicides due to their sensitivity. | [30,47,54] |
| **Acid and alkaline phosphatase** | *p*-nitrophenyl phosphate | Catalyzes the hydrolysis of organic esters, releasing phosphate and phosphoric acid anhydrides, and is considered an indicator of organic phosphate mineralization. Related to the amount of available OM. | [30,47,54,60,63] |
| **Arylsulfatase** | *p*-nitrophenyl sulfate | Catalyzes hydrolysis of aromatic sulfate esters in phenols and sulfate, is related to the amount of OM, with greater activity in surface layers under natural conditions. | [30,47,54,60,63] |
| **Ureasa** | urea use | Catalyzes the hydrolysis of non-peptide bonds, mineralizes N to $CO_2$ and $NH_3$, indicates losses of N in the form of ammonia. | [47,54] |
| **Nitrogenase** | | Participates in the reduction of nitrogen gas to ammonia and acetylene to ethylene (rapid sensitivity), measures nitrogenase activity, detects N fixatives (new symbiosis). | [25] |

## 4.3. Minimum Number of Indicators

In many cases, the limiting factor for measuring the soil quality through indicators is the cost, especially if biological parameters are included. Therefore, the total indicators contemplated to analyze the quality of the soil, at the initiative of several researchers, should be reduced to a minimum set of data [43].

The number of selected indicators usually varies between six and eight, and those that show the most relevant variations are chosen [43]. The selection and validation depends on the sensitivity

and response to climatic changes, as well as their accessibility (sampling) [58]. The objectives of the investigation are also considered in selecting indicators [43].

The criterion for the selection of biological quality indicators is the score awarded by recognized researchers, frequency of use, reproducibility (essential aspect), and topicality in publications [43]. Under the reference framework set forth in Sections 3.2 and 4.2, a possible set of indicators applicable to the soil context of the Ecuadorian Amazon is proposed in Table 6.

**Table 6.** Example of a minimum number of indicators of soil biological quality. Source: [43].

| Level | Indicator | Methodology | Principal Functions |
|---|---|---|---|
| **Population and community** | Presence, richness, and abundance of individual soil organisms | Traditional methods, microscopic, molecular techniques. | Cycle of OM and water, soil structure, regulation of microorganisms |
| | Microbial and fungal biomass | Plate count, fumigation, and extraction with chloroform | Cycling of OM and elements, soil structure, decomposition |
| | Indices based on soil biota communities | Identification and counting of the groups of organisms | Cycle of OM and elements, regulation of biological population, decomposition |
| | Community composition | Taxonomic identification and counting manual | Cycle of OM and elements, regulation of biological population, decomposition |
| **Ecosystem** | Soil respiration, nitrification, and denitrification | Evolution of $CO_2$, emission of $N_2O$, and production of $NO_3$ | OM and water cycling, decomposition, habitat provision |
| | Potentially mineralizable nitrogen | Anaerobic incubation | Natural fertilization |
| | Metabolic or microbial ratio | | |
| | DNA and protein synthesis | Incorporation of thymine and leucine into DNA | |
| | Enzymatic activity | Extraction and incubation of soil enzymes in various substrates | OM cycling, biological population regulation, decomposition |
| | Metabolomics and metaproteomics | Evaluation and quantification of metabolites and proteins in the soil | OM cycling, regulation of the biological population, soil structure, decomposition. |

### 4.4. Field Indicators

A project at the European level has developed simple and easy-to-use tools for farmers. The GROW Observatory project aims to provide services to citizens and non-profit science [66]. On the one hand, it allows for the measuring of soil parameters at high spatial resolution in large geographic areas. However, it has also opted for visual evaluations of soil in the field, a technique that is being implemented worldwide and is considered sensitive enough to assess the structure of a soil [1]. Most of the methods are based on observations of soil structure, and its relationship with crop productivity [66]. Other authors have affirmed that the visual evaluation of the soil is not sufficient to determine its state, the state of an ecosystem, or the services it provides [43]. They suggest that the indicators be preferably quantitative variables and propose the use of qualitative variables as valid and useful when there is no quantitative information, or when the costs of quantifiable parameters are high [38].

Studies in tropical soils of Venezuela have shown a strong relationship between visual evaluation scores, physical properties, and soil quality indicators measured in the laboratory [1]. Some variables taken into consideration in the visual evaluation are: (a) texture (tape method); (b) structure, by direct macromorphological observation and using a reference table (granular, laminar, or blocose); (c) depth of horizon (measured in the field); (d) color (Munsell table); (e) soil erosion (presence or absence of grooves); (f) slope (clinometer); and (g) height (GPS). Some authors have also proposed texture as an observable parameter in the field [27,67,68]. To understand the variation and to be able to relate the scores of the variables, they adjusted the data to a common numerical scale, facilitating interpretation.

However, the guiding approaches that are still provided by farmers or people who work the land strengthen knowledge through on-site practice (real time) [38]. They have the ability to measure the status of any agency or community. They manage to hold a discussion with researchers, relating practice to theory [43,58]. For example, the health of the soil can be determined from observations in the soil, plants, presence of animals, and water quality, and then related to laboratory analysis.

### 4.5. Relational Indicators Integrated Index

A correct evaluation of the soils takes into account the behavior and functionality of the organisms that inhabit it. Morphology, seasonality, and degree of sensitivity are also part of the evaluation process [7], showing the state of the soil. Some examples of relational indicators are shown in Table 7; they indicate sensitivity or adaptability depending on their densities.

**Table 7.** Examples of relational indicators.

| Relational Indicators (Ratio) | Description | Author |
|---|---|---|
| Oribatidos/Astigmados | Allows the prediction and evaluation of the degree of disturbance caused by the change of use in the ground. Based on densities (population), it expresses the ecological state of the edaphic environment and allows for the inference of the integral functioning of the ecosystem. Domination of astigmados indicates that the medium is altered and unstable. | [7] |
| Oribatidos/Prostigmados | Allow the evaluation of disturbances and state of the edaphic environment, like the previous relationship. If there is a dominance of prostitutes (indicator of aridity), the imbalance of soil communities is irreversible. | [7] |
| Mite/Collembola | Useful for determining the degree of disturbance. If the density of collembola is greater, it indicates fertility and stability of the soil (conserved ecosystem), whereas if there are mites, it would be necessary to identify the dominant group and the function in the ground. This relationship expresses the ecological state of the edaphic environment. | [7] |
| Earthworms/Termites | Earthworm dominance means conserved habitats, and termite prevalence means less conserved habitats, as they are considered opportunistic and resistant to induced disturbances. | [6] |
| CMB and NMB/COT and NT | Reflect that the MB is determined by the quantity and quality of the OM. An increase or decrease in the content of microbial C and N, will depend specifically on soil management. | [62] |
| MB/Enzymatic activities | In wooded soils, they indicate inactivity of microorganisms due to limited availability of C and N. This is with the exception of acid phosphatase. | [62] |
| Soil respiration/CMB | Indicates the proportion of turnover and importance of OC in the soil for a general improvement. | [58] |
| GMEa/nematode functionality ratio | Is a clear indicator of changes in soil quality, demonstrating sensitivity to heavy metals. | [63] |
| C/N | Low values of N indicate low quality humus. The presence of lignins and phenols may decrease the amount of C. | [53] |
| Particulate OM/OM | If it is positive, it is considered an important indicator of the rate of decomposition. | [53] |
| Particulate OM/soil respiration | Related to OM cycling and nutrient availability. It shows the relationship between N mineralization capacity, quality of plant residues, and soil respiration. | [53] |
| Potentially mineralizable N/soil respiration | Is related to disturbance and acidity of the soil that favors fungal growth. It shows the relationship between OM, N mineralizable potential, and edaphic respiration. | [53] |

### 4.6. Integrated Soil Quality Index

The Integrated Soil Quality Index is an integrated index based on a combination of indicators. This index clearly reflects the environmental quality of the soil and facilitates the comparison between different uses and management practices (same or different type of soil). In the countries of the Amazon, such as Brazil, Colombia, and Peru, its potential has been proven. It is obtained from the sum of three subscripts (physical, chemical, biological) of quality [44]. This index is developed in three

steps: first, there is the selection of appropriate indicators, based on accessibility, ease of measurement, and sensitivity; second, the selected indicators are scored (more is better, optimal value and less is better); third, the integrated quality index is developed using a linear or additive model, combining the score of the indicators.

According to the attributes of the soil and the score, the indicators are grouped into subscripts of soil quality. In each subscript, the indicators are valued by the number of times each score is reached. The subscripts are divided by the number of indicators they contemplate, to integrate them (sum of subscripts) and normalize the equation of the integrated index [44]. It is important to avoid underestimating soil disciplines.

## 5. Sustainability of Territories in the Ecuadorian Amazon

In Ecuador, the expansion of the agricultural frontier and deforestation for livestock purposes involves intervention and destruction of natural forests and virgin forests, causing socio-environmental impacts in the Amazon Region, in which 52% of land is forests and 17.5% of the area is used for agricultural activity [69]. With the emergence of livestock as a source of income for the population [37], a variety of perennial and annual pastures have been established as monocultures that are part of livestock systems adapted to the conditions of the region. These affect the Amazonian soils, because more than 50% of the livestock areas are in the process of degradation, with a break in the water balance of the basins and an increase in greenhouse gases [27,69,70]. Similar results have been reported for other tropical regions because of livestock and agricultural intensification [71].

The damage caused to the ground after deforestation varies according to the use that occurs later [27]. Converting deforested spaces into areas of agroforestry arrangement with tree species (coffee, cocoa, rubber) that represent management practices aligned with the potential use of the region means minimal damage to the soil [27,35]. The conditions of a natural forest are reproduced in terms of the interception of the drops, decrease in runoff, litter deposition, and control of the water erosion process. In contrast, the danger of erosion by crops, whether annual or perennial, is very significant at the beginning of the cycle, when the soil is left unprotected from its cover. In this sense, silvopastoral systems play a multifunctional role. Their adoption implies favorable changes in various components and agroecological processes [72]. The coverage offered by tree and litter species can activate soil biology, increase the level of OM, improve fertility and its optimization [35]. In addition, they can reduce soil degradation processes, such as water erosion, compaction, or waterlogging [27].

Based on the aforementioned factors, it is necessary to determine variables that contribute to improving soil quality and avoiding degradation processes from a systematic view, considering the soil resource as a fundamental component of farm-level production [27], with the implementation of sustainable or less aggressive management practices with the environment [47]. In this sense, the need arises to develop indices and indicators that measure, spatially and temporarily, the sustainability of a territory, and through permanent monitoring, encourage adequate and precise management of natural resources [73]. One of the main resources is land, which requires rational management through agricultural policies and practices aimed at forming sustainable agricultural systems [47].

The greater the knowledge of the wealth of resources, cultural diversity, and ecosystems that make up the Amazonian territory, the more the state will be a correct administration [27]. To meet the needs of the current population and improve their quality of life, economic, ecological, and social sustainability are required. Economic sustainability implies taking charge of global costs, such as the reproduction of nature and benefits, including the integral management of ecosystems. Ecological sustainability refers to the use of the natural system in an integral way, within the restrictions and potential of its homeostatic mechanisms. Social sustainability means a social orientation of production that uses and respects the identity of cultures and promotes the broadest participation of society in fundamental development decisions [74].

*5.1. From Environmental Indicators to Sustainability Indicators*

5.1.1. Environmental Indicators

According to the Organization for Economic Cooperation and Development (OECD), an Environmental Indicator is a variable that has been socially endowed with an added meaning derived from its own scientific configuration, in order to synthetically reflect a social concern, with respect to the environment, and insert it coherently in the decision-making process [75]. Environmental management and investment that is oriented towards the sustainable use, research, conservation, mitigation, and restoration of natural resources involve various topics, such as habits, anthropic modes of production and consumption, demand and use of natural resources, generation and use of solid and liquid waste, use of technologies and types of energy in the production of goods and services, and problems with pollutants that are part of climate change.

In Ecuador, the territorial approach of the current "National Development Plan 2017–2021" has tried to approach the reality of the value of environmental damage and the economy for social and environmental improvement [76]. The plan stipulates that Ecuador will fully assume the protection and guarantee of the rights of nature, responsible management of natural resources, protection of biodiversity and soil, and implementation of responses to climate change, which guarantees the good living of rural communities [76].

However, in the Ecuadorian Amazon, the application of sustainable territorial development policies and strategies has not advanced as expected [27]. The appropriation and mismanagement of natural resources is not the same in all communities, nor is the degree of impact due to environmental impacts; hence, ecological-distributive conflicts and environmental injustices arise [77]. These conflicts are valued in different languages. For example, affected communities may request internalization of externalities and monetary compensation, arguing that the natural environment has great ecological, landscape, and sacred value, or that resources are excluded from the market by international provisions that protect indigenous groups and nationalities. The point is that the monetary dimension is the key point in any dialogue or conflict that involves interests of large companies in natural areas of interest [77].

5.1.2. Sustainability Indicators

Sustainability indicators arise by expanding the role of environmental indicators, integrating the four dimensions associated with the concept of sustainable development [78].

The economic indicator refers to the physical and objective basis of the process of growth and maintenance of the stock of natural resources incorporated into productive activities, setting criteria for renewable natural resources, for which the utilization rate should be equivalent to that of resource recomposition. For non-renewable natural resources, the utilization rate must be equivalent to the replacement of the resource in the production process based on the time planned for its depletion.

The environmental indicator is related to the ability of nature to absorb and recompose to anthropogenic aggressions, making use of the above reasoning, where the rates of waste emission as a result of economic activity must equal those of regeneration, which are determined for the resilience of an ecosystem.

The social objective is to improve the quality of life of the population in countries with serious problems of inequality and social exclusion. The basic criteria should be: distributive justice, in the case of distribution of goods and services; and the universalization of coverage for global education, health, housing, and social security policies.

The policy is linked to the process of building citizenship and seeks to incorporate the population into the process of development and decision-making.

From the Ecological Economy [79], the concepts of weak sustainability and strong sustainability have been defined according to the assessment of ecological damage and resource depletion [77]. The sustainability indicators that are explained below correspond to them.

Monetary Indicators of Territorial Sustainability or Weak Indicators

Monetary indicators of territorial sustainability evaluate the percentage of income and expenses that could be considered true income and loss of assets, respectively, from the sale and acquisition of products that meet the needs of a region, with the aim of reaching agreed monetary valuations of amortization of natural resources and environmental services [80,81]. They include [74,80]:

- The gross domestic product (GDP) green, which indicates economic growth, taking into account environmental consequences;
- The GDP ecologically corrected, which takes into account the valuation of non-renewable reserves. In the case of an exhaustion of the resource, there is a replacement of the natural capital;
- The Sustainable Economic Welfare Index (IBES), which corrects conventional measures of private final consumption (expenditure), considering social and environmental factors;
- The Patrimonial Accounts, which exist in order to include environmental variables in the production matrix.

These indicators do not contemplate some issues, such as irreversible damage to the environmental system and its accumulation, alterations of natural resource reserves, the diversity of units in the environmental system, and arbitrary monetary valuations [79].

Biophysical Sustainability Indicators or Strong Indicators

Biophysical sustainability indicators are oriented to evaluate the impact of socio-economic activity on the environment. For this type of indicator, it is not easy to compare situations. Their objective is didactic, not research, and the information on how to calculate them is restricted [80]. Some of them are [81]:

- Ecological backpack, which indicates efficiency in the use of matter and energy per unit of product. Its purpose is to assess whether there is a dematerialization of the economy over time;
- The ecological footprint, which allows the estimation of the environmental deficit of a given territory, highlighting the impact that a human group has on ecosystems in relation to resource consumption and waste generation. This measure reports the dependence of a community on the functioning (productivity) of ecosystems, regardless of whether it is outside their domain. It allows the monitoring of the impact of human actions, and it should be noted that the agglomeration (cities) demonstrates the dependence on ecosystems and the environment [77];
- The environmental space, which refers to the quantity and availability of renewable and non-renewable natural resources in relation to the levels of contamination allowed without harming future generations' use of natural resources.

On a global level, the practical utility of these indicators in decision-making is limited. However, at the national, regional, or local level, they serve as instruments that can contribute to environmental planning and management [81].

*5.2. Territorial Sustainability Indicators: Participatory Selection*

The term "socio-environmental sustainability" refers to a dynamic balance between society and nature, as a result of a set of actions carried out with an integral vision of development processes. Part of the context and reality that it considers is the importance of not neglecting history, needs, conflicts, and potentialities. By analyzing the meaning of socio-environmental sustainability, it enables the creation of relationships within and between local communities, projecting an equitable use of resources in the short, medium, and long term [82], as well as the conservation of biodiversity, along with the participation of social actors. Within this context, territorial indicators allow the articulation of sustainability objectives and provide information on the state of the society–nature relationship. Its importance is based on the fact that it is formulated in a unique and unrepeatable context at a social,

administrative, and territorial level. The information and relationships established between selected variables will be driven by local managers [79]. The criteria to consider for the design of indicators are [81]:

- Political relevance, indicating aspects of collective interest and that are easily executed as concrete public actions;
- Feasibility that the analysis, collection, and processing is within reach of the community;
- Ease of interpretation; previous knowledge on the part of the actors to be understood and direct; the greater the value, the greater the sustainability;
- Validity; effectiveness of the purposes;
- Consistency with direct or indirect measurement (reality), without variations in time and space;
- Comparability; allowing decisions, differentiating situations, and establishing typologies;
- Synthetic, robust, or integrative; holistic as a parameter, inclusive in a small number of 59 reasonably added variables, with the possibility of disaggregation at local levels;
- Systemic; capable of integrating into a social monitoring system;
- Participatory; the allowance of the indicator to be involved in its own definition, analysis, and interpretation;
- Visionary or predictive; able to relate what is measured with community values, about a desirable future.

Any sustainability requirement is translated into an indicator [78,83]. Among the indicators proposed for the socio-environmental dimension and territorial sustainability, due to the impact of the establishment of productive activities that involve change of use and sustainable management of the soil, the Land Quality Indicators stand out [84]. The development and use of these indicators can influence the decision-making and planning of territorial development at the farm level or production units, to avoid the possibility that the rehabilitation of soil quality at a regional level does not show positive results. In this, the participatory selection of indicators is applied [27,84]. Soil quality is an integral part of achieving sustainable agriculture [30].

In the Amazon region, this perspective is intended to curb the impacts produced locally by deforestation, as a result of agricultural and livestock practices that expand day by day and influence the ability of soils to offer food security for rural populations [85]. Therefore, the proposed edaphological indicators are adapted to local needs. An example of the evaluation of sustainability using the methodology of participatory selection of indicators in production units according to four dimensions is summarized in Table A1 of the Appendix A. The example corresponds to the case of the Ecuadorian Amazon [69], and includes economic, cultural, and political aspects, since agricultural production also depends on them.

Direct measurements and observations of morphological parameters at the field level are a very useful tool. They allow an understanding of the operation and performance of the soil in the natural environment, and how it is affected by human intervention [60,69]. At the same time, they allow tracking or monitoring. Several authors [52,69,86] have selected the following: the morphology of the soil with observable attributes in the field (in surface or in the profile), the conditions of the environment, the behavior and development of the plants, and the response of the soil to management practices or changes in the use of the earth. The attributes described above include composition, soil structure, soil organization, color, root distribution, porosity, evidence of carbonates and iron, and clay and soil consistency. Other authors also proposed texture as an observable attribute in the field [27,67,68], as the OM content is based on the amount of litter and fresh and decomposing plant debris. This pair of parameters contributes in large proportions to the availability of nutrients and good physical and biological condition. Another main ecosystem service is the carbon storage potential [27]. In addition, it has been corroborated that the scores generated by the visual method in the field maintain a close relationship with those obtained in the laboratory based on quality indicators.

### 5.3. Accessible Space–Time Representations. Biography and Integrated Sustainable Development Index

For decision-making regarding sustainable territorial development, all types of information, data, experiences, or knowledge are raised from the highest level (national level) to the community and individual levels. A trend analysis is the best option obtain a (future) view of certain variables and indicators. Indicators that focus on sustainable development, and that contribute to self-regulated sustainability, serve as a solid basis for decision-making at each level [73], giving way to the development of an instrument of didactic work. Through an image, the degree of sustainable development of an area or unit of analysis (territory where strategies, policies, and investments will be implemented) can be represented and estimated, symbolizing the state of sustainability. An instrument called Biogram consists of a web image and the integrated sustainable development index.

The Biogram is a representation of a multidimensional diagram and indexes. It represents the performance of the unit of analysis, through the use of selected indicators, which allow overestimation or underestimation of the degree (development status) of sustainable development [73], on a chart. At the same time, it makes it possible to carry out a comparative analysis between the different dimensions and the possible existing conflicts. It results from the complement between the graph and the integrated sustainable development index, which allow for a quantification of the performance of the unit of analysis. To understand the information easily, the methodology standardizes or transforms the value of the indicators to a common scale (between 0 and 1), facilitating comparative analysis. In the spider web image (Figure 3a), each radius (axis) represents a calculation indicator. By methodological definition, the value of each indicator will vary between 0 and 1, with the minimum and maximum performance, respectively. Therefore, the greater the shaded area, the greater the performance of the unit of analysis.

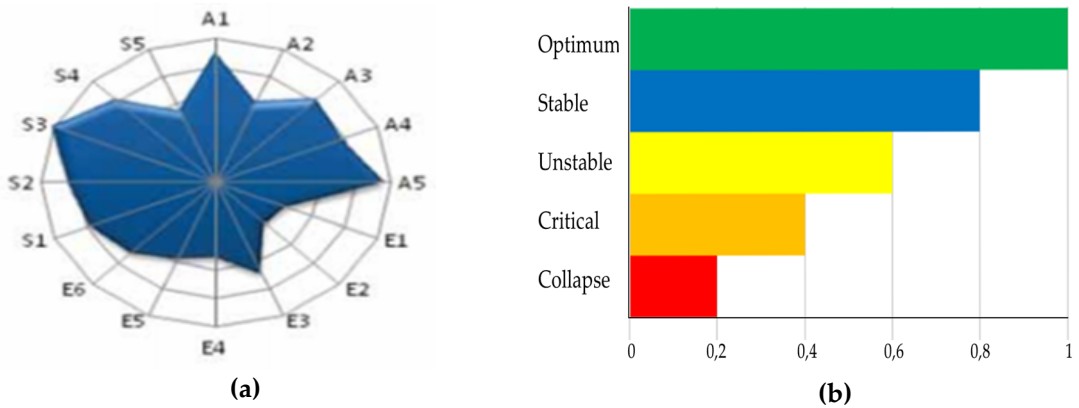

(a) (b)

**Figure 3.** (**a**) Biogram and (**b**) system status representation, based on [73].

The graphic representation provides knowledge on the individual performance of each dimension, showing an approximation of the degree of stability, and at the same time comparing the performance between units. The possibility of identifying the imbalances of an analysis unit in the image allows the location of the dimensions that need new policies or investments to correct the situation [73]. The Biogram uses five colors (Figure 3b) to characterize the state of sustainable development of the analysis units. When the shading red, it is less than 0.2, and it indicates a high probability of system collapse. Levels between 0.2 and 0.4 are represented by an orange color, and indicate a critical situation. Yellow shading, corresponding to levels of 0.4 to 0.6, indicates system instability. Levels between 0.6 and 0.8, shown in blue, represent system stability. The green color, for levels of 0.8 to 1, is the optimal system situation [73].

The methodology, on the one hand, allows for analyzing the evolution of an analysis unit in two moments (different years), and on the other, for comparing the situation of two units for the same moment (different places). One of the advantages of using this methodology is that it adapts to the

characteristics of territories based on the needs and peculiarities. From the score of the indicators, information can be obtained on the progress of the proposed objectives (social, economic, environmental, and political). It shows a synthesis of the reality of the territories in a simple but understandable way for the population.

## 6. Conclusions

In conclusion, this work has clearly highlighted the importance of the study of the biological component of Amazonian soils. We have developed tables that include specific indicators from the biological point of view. In addition, we showed the available methods for assessing the sustainability of Amazonian territories through the analysis of soil quality. Our contribution facilitates the need for the edaphic perspective to be taken into account in decision-making processes for sustainable territorial development.

**Supplementary Materials:** The following are available online at http://www.mdpi.com/2071-1050/12/7/3007/s1.

**Author Contributions:** Conceptualization, J.L.-M.; methodology, T.R.-T.; data curation, T.H.-L.; writing—original draft preparation, T.H.-L. & J.B.-S.; writing—review and editing, T.H.-L. & J.B.-S.; supervision, J.L.-M.; project administration, J.L.-M.; funding acquisition, T.R.-T. All authors have read and agreed to the published version of the manuscript.

**Funding:** This research was funded by "Ayudas para la realización de actividades de investigación y desarrollo tecnológico, de divulgación y de transferencia de conocimiento por los grupos de investigación de Extremadura" Junta de Extremadura, Fondo Europeo de Desarrollo Regional. GR-18169. Becas Internacionales Master Alianza Convocatoria 2019–2020, Junta de Extremadura—Universidad de Extremadura.

**Acknowledgments:** Isidro González Calatrava, Biblioteca Central de la Universidad de Extremadura, Robinson Herrera-Feijoo.

**Conflicts of Interest:** The authors declare no conflict of interest

## Appendix A

An evaluation of socio-environmental sustainability, and a proposal for a demonstration case selected in the Ecuadorian Amazon [69], with the indicators applied, are summarized in Table A1. The study concludes that, at the level of production units, each dimension presents critical levels, but with more economic and political limitations. According to the authors, the indicators that should be improved are the number of agroecological practices, crop diversification, productive diversification, management capacity, production level, net income, input costs, labor cost, marketing strategies, marketing strategy savings, production financing, institutional support, and state programs related to production units.

**Table A1.** Sustainability evaluation through indicators in production units of the Province of Napo, Ecuadorian Amazonia, based on [69].

| Indicators Applied to the Production Units (UP) with Their Respective Score Level, in the Province of Napo, Ecuador | | |
|---|---|---|
| **Dimension** | **Indicators** | **Optimum Level** |
| Environmental | Number of agroecological practices used | At least five must be agroecological, with ecological principles, such as association of crops, polycultures, cover, silvopastoral, and agroforestry systems |
| | Crop diversification | At least four types of crops |
| | Productive diversification | At least four productive activities |
| | Integral fertility: | |
| | - Texture | Franco, clay loam |
| | - Soil structure | Granular |
| | - Soil color | Dark brown or black |
| | - Apparent density | Less than 1.2 $Mg \cdot m^{-3}$ |
| | - Total porosity | Greater than 50% |
| | - pH | 5.5 to 7 |
| | - Organic material (%) | Greater than 3% |
| | - Nutrient availability (N, P, K, Ca, Mg) | Medium to high content |
| | - Biological activity | Abundant (wildlife is displayed on the ground) |
| | - Presence of erosion | Protected soil, without cracking |
| Socio-cultural | Number of people incorporated to the UP | All family members have active participation |
| | Workforce | At least 60% are familiar |
| | Task Distribution | Agreements are distributed equally |
| | Organization and management capacity | The UP is able to solve internal problems, keeps track of expenses, production and sales |
| | UP external administrative unit | Low, based on self-management |
| | Sense of belonging to the UP | High |
| | Acceptance of new practices | Willing to agroecological practices |
| | Waste disposal, recycling | Focusses on a specific area, apply recycling |
| | Time of dedication to the UP | More than 60% |
| Economic | UP production level | Self-supply and sale of surplus |
| | Acceptance of products (quality, price) | High |
| | Monthly net income in the UP | More than four minimum wages |
| | Agricultural input costs | 70% of inputs are produced in the UP |
| | Labor costs | 70% are members of the UP |
| | Marketing strategies | It has at least two strategies that give added value to the product |
| | Savings strategy | At least one strategy for eventualities |
| | Production financing | The unit is self-financing |
| Politics | State programs related to the UP (Chakra, livestock) | More than three programs associated with the UP |
| | Acceptance level of state programs | At least apply two programs or benefit from one |
| | Institutional support to the UP | At least three institutions supporting the UP |
| | Development of new skills and knowledge for productive work in the UP | All members have acquired new skills and knowledge, through courses and practical process workshops |
| | Level of participation or governance and associativity | UP members attend calls, technical tables, workshops, there is a predisposition |

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
