# Peer review of "A Framework to Incorporate Biological Soil Quality Indicators into Assessing the Sustainability of Territories in the Ecuadorian Amazon"

_sustainability, doi:10.3390/su12073007_

Round 1

Reviewer 1 Report

This paper states in the Abstract and Introduction that is about developing soil biological indicators of healthy, thus sustainable, function. The paper is very long and lacks focus. It presents as much information on native vegetation systems, agricultural production systems as on soil biology. Section 5 goes even more broadly into general sustainability indicators such as economic ones. A comprehensive sustainability 'biogram' is presented. A lot of the information is general and I never really get a sense from the paper of how soil biological indicators are useful to asses degraded land systems in Ecuador.

Maybe this should be broken into two papers?

Author Response

This paper states in the Abstract and Introduction that is about developing soil biological indicators of healthy, thus sustainable, function. The paper is very long (we are aware of the length and we have deliberated on how to approach this challenge. We have finally picked up this structure for several reasons.  We think that to be really useful, this review has to be that complete, not cutting it in two or three papers. But we are not unaware of the extension factor.  In any case, we have verified that other publications of Sustainability also have this  number of page. They are always in the REVIEW section, but our work is a bibliographic review, so it would fit into this typology) and lacks focus (the focus is the objective of the paper, expressed these terms (cf. lines 32-33) " compiling protocols and proposals for practical utilization”.

It presents as much information on native vegetation systems, agricultural production systems as on soil biology. Section 5 goes even more broadly into general sustainability indicators such as economic ones. A comprehensive sustainability 'biogram' is presented.

We agree and really appreciate this synthetic way of expressing the content of our work, and we are grateful to the reviewers for the time spent examining it. But here, really, we have nothing to discuss, we assent this has been the bulk of our contribution.

A lot of the information is general.

This is the main strength of this work. For this reason  it will be so useful. Its target audience is broad within the scientific community.  This transversal perspective is very necessary to build up multidisciplinary teams. We are lacking these sort of paper in Latin America.

and I never really get a sense from the paper of how soil biological indicators are useful to asses degraded land systems in Ecuador.

We understand this objection, mainly after having made all the changes asked by reviewer 3. All his/her modifications as well as those of reviewer 2, have resulted in a presentation of the tables that is certainly much better than our original manuscript. The general text of the document has been modified.  We welcome these drawbacks because they have certainly provided a more understandable perspective.

 Maybe this should be broken into two papers?

As we exposed in the abovementioned arguments, the option of maintaining the content in one paper is based on it is practical usefulness.

Reviewer 2 Report

This manuscript presents an overview of biological, edaphic and sociological measures to assess the sustainability of different production practices in the Ecuadorian Amazon. As the title suggests, the target audience of the manuscript is fairly narrow. Having said that, this article may still be of broader interest as a template for incorporating soil factors into sustainability assessments in other regions. And, given the fairly narrow target audience, I didn’t have any major concerns with the manuscript, as the authors did a commendable job of presenting their recommendations in the context of this specific region. I do have a couple of minor suggestions, though, which are listed below. 

p. 5, second paragraph: the second-to-last sentence suggests that the Ecuadorian Amazon is not suitable for agriculture, but the authors reasonably argue in the next paragraph that silvopastoral agriculture practices might be appropriate. If so, I would change this sentence to read ‘…not suitable for intensive agriculture…’

page 6, paragraph subtitled ‘Forests’: the first sentence uses em dashes followed by a full stop (period), which makes the em dashes seem inappropriate; a comma or rephrasing might work better instead. 

Also, first person singular (‘I’) is used here, while basically the rest of the manuscript appropriately uses ‘we’. There are a series of other typos (e.g., page 6, 4.1. Quality concept: the first sentence has an extra ‘it’ at the end of the sentence) scattered throughout the manuscript that should be correctable with another careful read through. 

page 12, Enzymatic activity. While it was fairly clear to me that Table 5 presents a representative list of enzymes that could be measured, rather than an exhaustive list of enzymes, the section and table heading might be rephrased to emphasise this point. 

Also, a couple of sentences in this section lack an object/noun. This issue occurs at other points in the manuscript as well; correcting these would help with the flow and citability of the manuscript. 

Author Response

This manuscript presents an overview of biological, edaphic and sociological measures to assess the sustainability of different production practices in the Ecuadorian Amazon. As the title suggests, the target audience of the manuscript is fairly narrow. Having said that, this article may still be of broader interest as a template for incorporating soil factors into sustainability assessments in other regions. And, given the fairly narrow target audience, I didn’t have any major concerns with the manuscript, as the authors did a commendable job of presenting their recommendations in the context of this specific region. I do have a couple of minor suggestions, though, which are listed below.

Thank you for your appreciations. Although we really have focused our work on Ecuadorian Amazon, our results are easily extrapolated to neighbouring countries such as Brazil, Peru, Colombia, Venezuela and the General Panamazon. The problem in these contexts is similar and we understand that our contribution is indeed a template for incorporating soil factors into sustainability monitoring. This aspect is very useful to implement the 2030 Sustainable Development Goals in that territories.

p. 5, second paragraph: the second-to-last sentence suggests that the Ecuadorian Amazon is not suitable for agriculture, but the authors reasonably argue in the next paragraph that silvopastoral agriculture practices might be appropriate. If so, I would change this sentence to read ‘…not suitable for intensive agriculture…’

Yes. We have changed it in line 105. Thank you, you are right.

page 6, paragraph subtitled ‘Forests’: the first sentence uses em dashes followed by a full stop (period), which makes the em dashes seem inappropriate; a comma or rephrasing might work better instead.

Ok we have added commas in line 157 and 158.

Also, first person singular (‘I’) is used here, while basically the rest of the manuscript appropriately uses ‘we’. It is a mistake.

It is true. It has been corrected in line 173   

There are a series of other typos (e.g., page 6, 4.1. Quality concept: the first sentence has an extra ‘it’ at the end of the sentence) scattered throughout the manuscript that should be correctable with another careful read through.

We have deleted it and corrected lines 174 and 175.  We would like  to communicate that we tried to do our best, by sending it to the Editing and Layout Service of MDPI, which certificate we uploaded to the platform. Although to be honest, unfortunately we had to write to the responsible person complaining about the service we had received on this occasion. This situation surprised us, because it was the first time since we are users of MDPI layout and Editing Service, that we have not obtained the normal high level quality of the service. In addition to this, for example all the Mendeley links of the references have been deleted, so (for example) we had to include the bibliography twice.

page 12, Enzymatic activity. While it was fairly clear to me that Table 5 presents a representative list of enzymes that could be measured, rather than an exhaustive list of enzymes, the section and table heading might be rephrased to emphasise this point.

Yes. We have changed as you suggest. We have written a new paragraph you can find from line 55 to line 62.

Also, a couple of sentences in this section lack an object/noun. This issue occurs at other points in the manuscript as well; correcting these would help with the flow and citability of the manuscript.

We accept your consideration and for this reason we have done our best to improve the redaction.

Reviewer 3 Report

The submitted text is to a lesser degree a stringent scientific evaluation. It is rather a compendium of soil quality issues. The text has the potential to be a useful reference paper for many purposes.

I recommend some improvements:

Table 1: in the column 'Function' there is a mix of terms. Not all of them are functions: examples are 'soil properties' and 'soil structure'. The authors may wish to be extremely accurate when defining the functions of organisms.

Table 3: as presently presented the table is not overly helpful. The authors give rather general indicators for the effect of organisms and support their statement with many references. When assessing the value of single statements it appears that the provided information is too general. Example: Earthworms - an indicator for the state of the soil! - Such a statement is not constructive. I recommend that the authors go through the table line by line and give a critical expert judgement under which circumstances some of the indicators work well. - The authors may also wish to translate all organism groups from Spanish to English.

Table 4 sees a similar critique as table 3. However, the table can stay as it is when the conditions in the experimental area are taken in consideration.

Chapter . 4.3 starts with the statement that more measured parameters are better than fewer parameters. This is not true. The value of indicator does not depend on the number but on their strength.

Table 6: first line: what is meant with 'wealth'? - would that eventually be richness?

Author Response

The submitted text is to a lesser degree a stringent scientific evaluation. It is rather a compendium of soil quality issues. The text has the potential to be a useful reference paper for many purposes.

Thanks ever so much you for your valoration. This is really the objective of our review paper:  to became a useful reference in multidisciplinary sectors related to environment in Ecuadorian amazon rainforest

I recommend some improvements:

       (We have tried scrupulously follow your appreciations, and we are really grateful to you for them. We have mentioned you  in acknowledgments, as “anonymous referee”, but if you do not mind to be mentioned, please do no hesitate to contact the Editor. )

Table 1: in the column 'Function' there is a mix of terms. Not all of them are functions: examples are 'soil properties' and 'soil structure'. The authors may wish to be extremely accurate when defining the functions of organisms.

This is really true. You are right. We have carefully review the Table, and we have introduced the changes mentioned below.

Line 72

  • We have added : They form a network of tunnels mixing and digging the soil, producing excreta below and above the ground, modifying the water and chemical properties [4]. They transform the physical properties of the soil, benefiting the formation of aggregates, the movement and retention of water, as well as the gas exchange [8]. They act as ecosystem engineers in pore formation, water infiltration and OM humification and mineralization [6][10].

And we have deleted: Decomposers, nutrient cycling, dispersers, soil properties, compound mineralization, and soil structure

  • We have added: They participate in the litter fractionation and in OM decomposition and mineralization processes [6]. The conservation of the coleoptera family can be a support for possible evaluations of the environmental quality [11].

And we have deleted: Decomposers, nutrient and OM recycling, soil properties, natural fertilizers

  • We have added: Its mounds are rich in nutrients such as N, P, K, Ca, Mg and Fe, favoring the proliferation of microflora and micromesofauna [12]. They affect the soil structure, mixing the horizons of the profile and recycling part of the elements that leach from the surface [13]. They modify the physical and chemical properties of the soil [10], and contribute to the formation of aggregates, water filtration and aeration [8].

And we have deleted: Nutrient cycling, dispersers, soil properties

  • We have added: They contribute to bioturbation processes [4],and intervene in the crushing of plant remains and in decomposition of woody material [6].

And we have deleted: Decomposers, nutrient cycling, soil properties

  • We have added: They participate in the litter fractionation and in OM decomposition and mineralization processes [6], and fragmentation of leaf litter, when they mobilize they secrete mucus, increasing area for microflora activity [4][8]. Its mucus helps aggregate formation, improving soil structure and properties [10].

And we have deleted: Decomposers, soil properties

  • We have added: They live among the leaf litter or under the bark of trees and rocks, they play an important role as predators, and others participate in leaf litter fragmentation, speeding up the OM decomposition process [8][10].

And we have deleted: Decomposers, predators

  • We have added: They participate in leaf litter maceration and plant remains, facilitating the transport of excavators, and they can also act as predators [10].

And we have deleted: Predators

  • We have added: They are decisive in the recycling of organic waste, dividing and crushing them, their excreta benefits the roots by the continuous release of nutrients. They participate as predators of nematodes and fungi [7]. Are considered a decisive element in the recycling of organic remains, and contribute to the structure of the soil [10].

And we have deleted: Decomposers, nutrient cycling, predators (regulators), soil properties, fungal disease control

  • We have added: Their role is to fragment leaves and dead wood, disperse microbial and fungal spores in the soil. Some species are predators of other microarthropods, nematodes, and mites [10]. Moreover, they contribute to the soil stability and fertility [7], and in OM decomposition [14].

And we have deleted: Decomposers, dispersers, predators, soil properties

  • We have added: They are concentrated in the roots, serve as food for plants, do not participate in OM decomposition [10], reflect the OM availability in different ecosystems, are the link in the food chain between microorganisms and complex organisms [7]. Some can resist soil disturbances and chemical pollutants [15], others are parasites. There are mycophagus, bacteriophages participate in the regulation of available nitrogen and phosphorus and influence the Rhizobium nodulation [10][14]. Are important agents of the nutrient cycle and regulators of soil fertility, and they work as biological control agents [15][16].

And we have deleted: Nutrient cycling, predators, fertility regulators, pathogens, biological control agents

  • We have added: They inhabit deep strata, under trunks or stones, they are detritivorous and depend on moderate and constant humidity, they consume microorganisms and fungal hyphae, which is why they are considered to be involved in decomposition, some of their representatives are predators and phytophagous [7][10].

And we have deleted: Decomposers, predators

  • We have added: They are considered the most important predators of bacteria and fungi. Moreover, they regulate microbial communities, and as pathogenic insects, represent an important biological control [4][10].

And we have deleted: Predators, pathogens

  • We have added: Are involved in processes of decomposition, mineralization, and cycling of nutrients [2][17]. By forming symbiotic associations, they increase the efficiency of plants to absorb nutrients [4], and increases soil aggregation and participates in the carbon cycle [17], and allow plants to survive and efficiently absorb phosphorus from the soil [18]. They improve the soil health and plant species growth, provide greater absorption of nutrients, uptake of immobile ions, tolerant to toxic metals, root pathogens and unfavorable conditions for plants in tropical ecosystems [2][10][19].

And we have deleted: Decomposers, soil structure, extension of the root system of plants, greater absorption of nutrients by association with mycorrhizae, pathogens, disease control in plants

  • We have added: They are photosynthesizing organisms involved in primary production, OC compounds, and soil structure [10], and are colonizers. In association with fungi, they form lichens and contribute to soil formation, degrading minerals or rocks by excreting organic acids [20]. From the production of carbohydrates, they form soil aggregates and stopping their erosion [21]. Given the variable and morphologically similar nature of the majority, today, they are identified using molecular techniques [22][23][24].

And we have deleted: Colonizers (primary production, OC), soil structure, food source (they are at the base of the food chain)

  • We have added: They rarely contribute to biological activity. They can be considered bags full of enzymes [10]. In the soil they are very numerous and genetically different. Some degrade chemical compounds and others form nodules in the roots of legumes, with the function of fixing atmospheric nitrogen through heterocysts. Cases such as Pseudomonas can be pathogenic [10][25]. There are cyanobacteria (photosynthesizers and autotrophs) [26]; Actinobacteria are colonies similar to fungal mycelia, like Actinomycetes that degrade OM to form humus and participate in the mineralization process, others can fix N or regulate the composition of the microbial community in the soil. They secrete enzymes that serve for the biological control of nematodes, insects and other soil organisms. Their number on agricultural land is high [2].

And we have deleted: Degradation of chemical compounds, fix atmospheric nitrogen, pathogens, mineralization process, biological control

Table 3: as presently presented the table is not overly helpful. The authors give rather general indicators for the effect of organisms and support their statement with many references. When assessing the value of single statements it appears that the provided information is too general. Example: Earthworms - an indicator for the state of the soil! - Such a statement is not constructive. I recommend that the authors go through the table line by line and give a critical expert judgement under which circumstances some of the indicators work well. - The authors may also wish to translate all organism groups from Spanish to English.

Thank you very very much for your critical comments on this Table, which really have make us to review its data from a new perspective. Following your considerations we have gone through it line by line, and we have introduced many detailed changes that have resulted as a real improvement of the quality of the summarized information.

Line 240

  • We have added: They are recognized for presenting sensitivity to anthropogenic disturbance, proposing them as indicators of soil degradation [4][13]. Several authors propose them as a biological indicator of the state of conservation/alteration of the soil according to the composition and abundance [6]. Their presence indicates preserved habitats [55].

And we have deleted: Indicator to assess the state of the soil

Soil disturbance bioindicators

Their presence indicates preserved habitats  [4][6][8][10][41][42]

 We have added: They are considered excellent bioindicators to evaluate anthropogenic intervention due to the high sensitivity to environmental variations and deterioration of ecosystems [6][40]. According to the ecological niche they occupy, they are considered as indicators of the conservation status of the ecosystem [39].

And we have deleted: Indicators of soil disturbance status

Bioindicators to evaluate anthropic intervention        [6][8][23][24][28][43]

  • We have added: The presence of termites indicates less conserved habitats or habitats with a certain level of degradation, considered opportunistic organisms due to resistance to induced disturbances [6]. On the other hand, they are potentially the most important taxa as ecological indicators, because they are at the ecological center of many tropical ecosystems [12]. Moreover, for its sensitivity to environmental or anthropogenic disturbances in biotic systems [56].

And we have deleted: Indicator of less conserved habitats      [6][10][40][44]

  • We have added: Used to indicate the state of disturbance in the edaphic environment, they are very sensitive to sudden changes in humidity and temperature, associated with vegetation cover and the entry of residues [6]. For this reason are considered indicators of humidity and temperature changes.

And we have deleted:  Indicators of humidity and temperature changes

Indicators of soil disturbance status               [4][6][8][10]

  • We have added: Used to indicate the state of disturbance in the edaphic environment, they are very sensitive to sudden changes in humidity and temperature, associated with vegetation cover and the entry of residues [6], and because these changes can influence its functions and abundance [8].

And we have deleted: Indicators of soil disturbance status      [6][8][10]

  • We have added: They are drought sensitive organisms [10], for this reason they are considered as drought indicators. And can be considered as bioindicators of soil stability and fertility [7].

And we have deleted: Drought indicators

Bioindicators of soil stability and fertility       [7][10]

  • We have added: Due to their action of reducing fungal concentrations, in crops they are used as bioindicators of soil contamination, since they have whitish and soft bodies, they are considered an indicator group of fertility and stability of the edaphic environment due to their sensitivity to chemical products and environmental disturbances [7]. For the changes in their composition, they are considered indicators of ecological variations, due to the influence of agricultural practices, making the presence of taxa effective as bioindicators of herbicide treatment [57].

And we have deleted: Indicator of fertility and stability of the edaphic environment

Compaction Indicators

Indicators of ecological variations

Bioindicators of soil contamination (herbicides)

Bioindicators of agricultural potential of soils              [7][10][45][46]

  • We have added: Due to their morphological and bioecological characteristics, they are very demanding in terms of habitat quality, proposing them as potential bioindicators of disturbance as they are sensitive to OM content, humidity, pH, agricultural practices, use of insecticides and environmental changes. They respond positively to good soil aeration conditions, considering them indicators of stable and productive soils. And in soils not intervened as bioindicators of low heavy metal values [7].

And we have deleted: Biological indicators of moisture and OM content

Compaction Indicators

High agricultural productivity indicators

Bioindicators for their sensitive to anthropic disturbance and environmental changes

Low-level heavy metal bioindicators             [7][10][46]

  • We have added: Surviving unfavorable environmental conditions, they are proposed as good indicators of disturbed soils [7].

And we have deleted: Disturbed soil indicator

Soil instability indicator     [7][10]

  • We have added: Biological indicators of soil stability and fertility, due to their susceptibility to environmental disturbances and the fragility of their whitish bodies, these characteristics also make them a good indicator of soil quality, since they are abundant in the least disturbed [7].

And we have deleted: Indicators of soil quality

Compaction Indicators

Sensitive to disturbances   [7][10]

  • We have added: When they have high dominance, it is considered as an indicator group of the aridity and the imbalance of the edaphic communities is irreversible, because they have a high reproductive potential, which allows them to adapt to the disturbance and for this reason they are considered disturbance indicators [7].

And we have deleted: Aridity Indicator

Good disturbance indicator to the environment due to their fast adaptability to disturbances               [7][10]

  • We have added: They act as biological control agents for pests and insects, qualifying them as powerful bioindicators of ecological conditions [15]. Through appropriate analysis of the nematode community, the level of contaminant disturbance and changes in land use can be estimated [43], therefore, they are considered as indicators of sensitivity and stability indicators [16].

And we have deleted: Soil stability indicators

Indicators of environmental disturbances due to their sensitivity

Bioindicators of ecological conditions         [7][10][27][47][48]

  • We have added: Due to their morphology and trophic functions, they are considered as indicators, they are very sensitive to agricultural practices, thereby reducing their population [7].

And we have deleted: Soil stability indicators

Sensitive to agricultural practices    [7][10]

  • We have added: The mycorrhizal association has recently been seen as an important indicator to assess soil quality. They also represent a key group of organisms in the soil that can affect plant productivity, biodiversity and characteristics related to ecosystem sustainability [17]. Moreover, are considered as bioindicators of soils contaminated by heavy metals [58].

And we have deleted:  Important indicator to assess soil quality

Bioindicators of soils contaminated by heavy metals  [2][4][10][35][49][50][51]

  • We have added: Excretions of fatty acids and carbohydrates, stop erosion forming aggregates [10]. Due to its nature and similar morphology, molecular techniques are used for better identification [22][24].

And we have deleted: Excretions of fatty acids and carbohydrates, stop erosion forming aggregates            [10][52][53][54][55] [56]

  • We have added: The actinomycetes in tropical soils are one of the most important bacterial groups [2], as indicated by a recent review of soil bacteria worldwide [59]. Its systematic classification is based on molecular techniques (16S rRNA sequencing) of soil microorganisms [26][59].

And we have deleted: Considered bags full of enzymes, nitrogen fixers

Used as chemical compound degraders and microbial regulators           [10][57][58][59]

Table 4 sees a similar critique as table 3. However, the table can stay as it is when the conditions in the experimental area are taken in consideration.

We have introduced significant changes in Table 4 in order to improve its quality. We hope now we have managed to present an improved version of it. The following modifications have been made.

Line 247

  • We have added: Is considered as important indicator of soil quality and productivity [30], because it influences a wide rang of soil properties. On the other hand is the most important component of soils, since it plays a key role in determining physical, chemical and biological processes, exercising crop production [47]. OM is globally recognized variable as the universal indicator of soil quality [53].

It is considered an sensitive indicator to changes due to soil management [58][60].

And we have deleted: Indicator of biological quality and agronomic sustainability

Important indicator of soil quality and productivity

Indicator sensitive to changes due to soil management             [14][31][38][51][60]

  • We have added: Both are positioned within the most sensitive indicators, helping to identify changes manifested at different depths and in the face of management practices [53]. Organic phosphorus allows predicting nutrient availability in short term [19].

And we have deleted: Allow prediction of short-term nutrient availability

Indicators very sensitive to anthropic intervention     [38][35][38]

  • We have added: It is a necessary indicator for a complete evaluation of the soil, it is associated with the quality of the OM. on the other hand, it corresponds to the amount of organic soil nitrogen that can be converted by microbial activity to soluble inorganic forms and due to its sensitivity it can be used as an indicator of the production capacity of the soil or as an indicator of the nitrogen contribution of the soil to support recommendations for the application of N [53]. Moreover, as an indicator highly sensitive to changes in use in subtropical soils.

And we have deleted: By its sensitivity, it indicates the nitrogen contribution of the soil

Indicator for assessing soil quality

Indicator sensitive to changes in use in tropical soils  [38]

  • We have added: Microbial biomass is considered an indicator of soil fertility and quality, and is negatively affected by changes in land use and agricultural practices [60]. Furthermore, they are used as indicators of the first environmental changes by deforestation.

Microbiological parameters that have been used as an indicator of the effect of agricultural practices and pollutants on soil quality [47], and as indicators of the relationship between biota and the restoration of degraded systems, allowing us to know the abundance and population structure of microorganisms [58][62].

They have been proposed as indicators of soil quality in natural and agricultural systems, due to the role of microorganisms in the C, N cycle [61], and their sensitivity [30]. In subtropical soils they can serve as potential biological indicators of ecological changes resulting from land use and management practices [61].

And we have deleted: Important ecological indicator.

Soil fertility indicator

Low levels indicate agricultural impact

Indicates soil quality and allows estimating the population structure of microorganisms

Potential bioindicators of ecological changes

Sensitive indicators of changes in land use   [31][51][60][61][62][31][51][61][62][14][51][61][62]

  • We have added: It is carried out by microorganisms under aerobic conditions, it is a useful index to know the amount of easily mineralizable substrate, by determining the amount of CO2 released by the action of biological activity and the easily mineralizable OC [47][54]. Allowing to evaluate the type of management to which a soil is subjected [47].

Indicator highly sensitive to changes in the use of subtropical soils [53].

Used as an indicator of soil quality and microbial activity, it can also indicate the amount of easily mineralizable substrates [63].

And we have deleted: Used as an indicator of biological activity

Indicates quantity of easily mineralizable substrates

Indicator sensitive to changes in use in tropical soils  [31][38][39][63]

  • We have added: The metabolic processes that occur in the soil can serve as early and sensitive indicators against the changes caused by differents soil management [53].

This quotient is a useful indicator to monitor changes in OM and is often used as a sensitive index to measure changes in soil OC. Its increase is considered as an indicator of environmental stress after the conversion of forests to farmland [60].

Indicator of availability and quality of microbes, it is also sensitive to other factors such as the proportion of fungal and bacterial biomass [63].

The stress of the microbial communities can be quantified by means of this parameter that reflects the energy requirement or indicates a change in the bacterial-fungal ratio. At the same time, it could be a useful parameter in the study of bioenergetic changes in developing ecosystems [64].

And we have deleted: Indicator of availability and quality of microbes

High content indicates disturbances and microbial stress         [38][60][63][64]

  • We have added: It is a measure of enzyme activity that is proposed as an indicator of recovery in the presence of bioavailable heavy metals [54].

It is an indicator of changes in soil quality under different agricultural management practices and is used to assess the effects of cultivation on soil quality [60].

It has been shown to be a good index (condensing the set of enzyme values) to estimate the quality of the soil, since it is related to other physicochemical or biological properties of the soil. Furthermore, it is an early indicator of change in soil quality and is sensitive to metal contamination [63].

And we have deleted: Good index to estimate soil quality       [39][60][63]

  • We have added: The estimation by direct or indirect extraction of nucleic acids (DNA and RNA) from the soil and their subsequent study through molecular biology techniques such as PCR, has been used successfully, and they have been proposed as indicators of microbial biomass active [47]. This technique is capable of discriminating between bacterial and fungal biomass.

they are the preferred way to assess the structure and dynamics of the soil microbiological community, since microbiological life is a favorable indicator of adequate soil characteristics [65].

And we have deleted: Indicators of active microbial biodiversity

Indicate the relationship quality / functionality of the soil

Indicates damage to ecosystems by agricultural practices

Indicate soil sustainability [31][65]

  • We have added: They are considered as sensitive indicators between reforestation methods and changes produced by agricultural activity [30].

Due to their sensitivity, relationship with biological activity and rapid response to changes that occur in the soil (use and management), they have been proposed as potential indicators of soil quality [47][61]. Phosphatase activity is a soil indicator to estimate the potential mineralization of organic phosphate [47].

They are used as indicators of changes in soil microbial activity, in response to heavy metals and corrective measures [54], and they show sensitivity to changes in land use and pollution.

They have been used as indicators of soil fertility and quality, but while being negatively affected by changes in land use and agricultural practices are proposed as indicators of soil degradation [60][63]. They can also indicate accelerated decomposition [63].

And we have deleted: They show sensitivity to changes in land use and pollution indicating the degree of affectation.

Indicates accelerated decomposition

Indicates ecological changes resulting from land use

Indicates goats in biological properties of tropical soils Catalyze biochemical reactions essential for the life of organisms.

Indicates fertility and soil quality    [14][31][39][60][61][63]

Chapter . 4.3 starts with the statement that more measured parameters are better than fewer parameters. This is not true. The value of indicator does not depend on the number but on their strength. 

Thank you for this consideration because the sentence produces misunderstanding. We have deleted lines 72-73

Table 6: first line: what is meant with 'wealth'? - would that eventually be richness?

Yes it is a mistake. It has been changed by richness. (cf. after line 85)

Round 2

Reviewer 1 Report

The authors have made some revisions but I still don't see how the biological indicators have been applied to the case study. Maybe this because there isn't much data for Ecuador on all organisms listed in Table 1.  

A change of title is needed "A framework to incorporate biological indicators into ..."

Author Response

Here we enclose the revised version of the manuscript sustainability-737171 and our response (in red)  to the Reviewer #1  Report (Round Two) : 

 The authors have made some revisions but I still don't see how the biological indicators have been applied to the case study.  Honestly we think they have been clearly proposed, as to be used at any level. At the fields, with a minimum number of Indicators proposal, and through the most sophisticated biotechnological PCR and molecular techniques of identification available only at High Level Research Departments.

 Maybe this because there isn't much data for Ecuador on all organisms listed in Table 1. If you carefully consider the composition of elements of Table 1, you can observe how the selection of the information that has been incorporated has been made taken into account the amazonic perspective. In Spain, or in North Canada or Africa for example it should have a different perspective..  

A change of title is needed "A framework to incorporate biological indicators into ..." THANK YOU very much for this suggested change, which really improves the title of our work. We have modified the title of the paper  following your indications as it can be seen in the final version of the manuscript. “A Framework to Incorporate Biological Soil Quality Indicators into Assessing Sustainability of Territories in the Ecuadorian Amazon” is a new title.

We are very grateful for your time and consideration

Best regards,